# Which Factors Influence Attentional Functions? Attention Assessed by KiTAP in 105 6-to-10-Year-Old Children

**DOI:** 10.3390/bs9010007

**Published:** 2019-01-08

**Authors:** Marta Tremolada, Livia Taverna, Sabrina Bonichini

**Affiliations:** 1Department of Developmental and Social Psychology, University of Padua; Via Venezia, 8 35131-Padova, Italy; s.bonichini@unipd.it; 2Faculty of Education, Free University of Bozen-Bolzano, Brixen-Bressanone; Viale Ratisbona, 16 39042-Bressanone, Italy; livia.taverna@unibz.it

**Keywords:** attentional functions, primary school, KiTAP, healthy children, gender, delays

## Abstract

This research revealed the children with difficulties in attentional functions among healthy children attending primary school and aimed to identify the possible sociodemographic factors, such as the child’s age, gender, and school grade, that could influence attentive performance. The participants were 105 children aged 6–10 years (M age = 8.6; *SD* = 1.04), attending primary schools. Family economic condition was mostly at a medium level (63.5%), and parents most frequently had 13 years of schooling. The computerized test KiTAP was administered to children to assess their attentional functions. Results showed a higher frequency of omissions and false alarms and a reduced speed in alertness, go/no-go, and sustained attention tasks compared to Italian norms. Hierarchical regression analyses were run with school grade, gender, and current age as independent variables and mean reaction times (and standard deviation), number of omissions, and false alarms as dependent ones. The results showed that male gender and attending a lower grade impacted on lower attentional performance in several subtests. Girls showed the best performances in tests of distractibility and impulsive reaction tendencies, while higher school grade positively influenced divided and sustained attention. These results could be useful to identify children with major attentional difficulties, and some recommendations for future studies and the implementation of attention empowerment programmes are proposed.

## 1. Introduction

### 1.1. Definition of Attention and Adopted Theoretical Model

Attention has been identified as a complex construct in psychology which does not express a unitary concept but concerns a psychological phenomenon that interacts with all other cognitive processes, such as perception, memory, behavioral planning or actions, linguistic production, and spatial orientation [1]. Attentional skills are a prerequisite for responding to daily environmental demands in that, through them, a person can select and integrate all the relevant information he/she perceives, coming from different sensory channels, and associate them with conceptually superior categories. Besides cognitive processes, motivational and emotional processes have been recognized as also having an important role: what is perceived as not interesting, without an affective value, does not become a subject of attention [2]. Attention involves several developmental tasks, including focused attention, sustained attention, attention shifting, and divided attention. Inhibitory control is commonly described as the ability to suppress a dominant or automatic response: response inhibition [3]. It encompasses an attentional component known as interference control: the ability to selectively attend to certain stimuli and ignore irrelevant stimuli [4]. Focused attention refers to being able to actively focus on one thing without being distracted by other stimuli; sustained attention can be defined as the ability to maintain concentrated attention over prolonged periods of time [5].

In the present study on attention in primary school children, we adopted the aspects of attention model proposed by Van Zomeren and Brouwer [6]. Attention can be categorized into three main components depending on their different functions: (a) activation (alertness, sustained attention), (b) visual–spatial orientation (overt attention, visual search), and (c) selective executive components (divided attention, inhibitory control, and flexibility). The two authors schematized the basic processes of attention by grouping them into two main reduced components: selectivity and intensity. Within selectivity, they distinguished focused attention and divided attention, while alertness and sustained attention (or vigilance) were incorporated into intensity (Figure 1).

The development of these functions is generally accompanied by the neurological maturation of particular areas in the brain and can be clearly distinguished by these areas’ development pathways. The parietal and back cortex (visual or spatial attention) are involved in the basic attention processes, before the main executive functions, through automatic inhibitory activity, suppress irrelevant information. Executive functions have been associated with the slower maturation of the frontal and prefrontal cortex areas [8,9,10,11,12]. This implies that chronological age has an essential influence on attention performance during development, but also that academic achievement allows pupils to train their attentive skills. 

A specific test to assess attentional performance for children (KiTAP) has been developed. The KiTAP is a standardized tool with exceptional psychometric properties and has been used in recent neuropsychological research on children with neuropsychiatric disorders such as Attention Deficit and Hyperactivity Disorder (ADHD) [13], as well as in children who received a liver transplant [14] and children with motor coordination impairments [15]. The procedure is based on the quantitative and qualitative features of Van Zomeren and Brouwer’s neuropsychological model [6]. 

### 1.2. Factors Influencing Attention Performance in a School Context

The strong interindividual variability in attention performance depends on a number of factors, both constitutional and environmental, that determine the different developmental paths that attention could follow. Thus, as with all cognitive skills which are developed, in order to be understood and evaluated as fully as possible, consideration should be given to the child’s characteristics, taking into account the influence of many factors [16]: the biological characteristics of the child (i.e., temperamental characteristics favouring girls for effortful control and boys for surgency [17], maturation levels of the central nervous system (activation and visual spatial attention that show earlier development than other executive functions [8], general cognitive and emotional capacity of the child [2], and environmental variables, namely his/her personal experiences and the context in which he/she lives (for example, supporting parenting strategies) [18]. As far as environmental variables are concerned, we take into consideration the primary school experience because this period is characterized by rapid changes in attention functions according to the literature, and thus the role of attention in academic learning and achievement may be critical [19,20]. Scholastic achievement is positively correlated with attention-related skills and the development of attentional processes. However, in most investigations, attentional performance has been assessed using teacher and parent ratings of children’s ability to focus and shift attention, thereby introducing a risk of rater bias [21]. While there appears to be an association between attentional processes and scholastic performance, the specific aspects of attentional performance that are associated with scholastic achievement are unknown. School is one of the most significant and privileged developmental contexts for the child. With his or her attendance of primary school, the child faces new developmental challenges compared to early childhood, which will lead him/her to an important cognitive, emotional, and social evolution [22]. The class that the child attends, people around him/her, and everything defining the child in his/her specificity assume an important role in attention performance. 

Ample evidence suggests that children’s executive functions can be improved by interventions [23], schooling [24], and environmental factors [25] during that time. For this reason, this study aims to directly assess attentional functions in children within the school environment, taking into consideration not only the chronological age but also the child’s academic achievement. The results of a study by McCrea et al. [24] demonstrate that formal schooling provides small-to-moderately sized effects on the development of executive function during the early school grades and that this effect is highly dependent upon the nature of the task. A large schooling effect on the verbal fluency measure at age 8 was detected, suggesting that schooling may directly improve performance on executive measures. Such schooling effects could conceivably arise from several factors, including an increased knowledge base, learning of general problem-solving strategies, or the learning of domain-specific strategies [26,27].

#### 1.2.1. Role of Age

Several studies on children’s attentive development have shown that the most rapid improvements in alertness, sustained attention, and spatial orienting (visual search) occur between the ages of 6 and 10 years [28,29]. With respect to the development of executive functions such as inhibition, flexibility, and divided attention, there is growing evidence that these functions actually emerge in the early years of childhood, but that they still develop gradually into late childhood and early adolescence [10,30]. Attentional functions, like all the cognitive mechanisms, are primarily affected by the level of cerebral maturation. Throughout childhood and until adolescence, the so-called “executive” attention will be defined to control behaviours, to distribute cognitive resources, and to plan and direct action to achieve specific goals [31]. This could not happen if the central nervous system and the targeted networks did not mature. 

Zimmermann and Fimm [32] studied the general development of attention in healthy children aged 6 to 12 years. Despite the unavoidable interindividual differences, they observed that increasing age inevitably increased the quality of performance on attentional tests and that these performance levels, initially very heterogeneous, tended to stabilize. Reaction times, for example, very different in children aged 6–7 years, decreased as their age increased and seemed to stabilize only at the age of 13–14 years. Flexibility, important to control the focus of attention, also grew with child maturation. In addition, the results for tests of split attention showed how it was influenced by age. The influence of age was more evident on performance speed than on its quality in 5-to 11-year-old Arab children, with rapid improvement until the age of 9 years, with some attentional functions (alertness and inhibitory control) that seemed to develop earlier than other functions (distractibility and divided attention) [33]. Age was negatively associated with distractibility, lapses of attention, and cognitive speed, indicating that these parameters decrease with age in healthy children [21]. The number of errors (incorrect responses to critical stimuli) and omissions (missed responses to critical stimuli) were found to be critical attention scores for academic performance in primary school children and seemed to constitute a sensitive measure of distraction [32].

Inhibitory control shows rapid development during the preschool years but also improves between 5 and 8 years of age [3]. Early attentional control development peaks during the preschool years, although it continues to develop during the primary school period alongside the emergence of the core executive functional components [34].

The results of a recent study [35] indicate significant age-related improvements between 8 and 10 years in all the attentional functions, particularly regarding the developmental rates for divided attention, sustained attention, and flexibility.

In this cross-sectional study for the first time in the Italian population, it is important to confirm or disconfirm these possible child age differences in developing attentional functions by inserting this variable into a predictive model that could weigh the best predictive variables, including the child’s age. 

#### 1.2.2. Role of Gender

The literature offers a wide range of studies, for example Biedereman et al. [36] and Siegel and Smythe [37], that have investigated the most influential factors in attention development according to gender, but only in cases of disorder or pathology. For example, it was noted that Attention Deficit and Hyperactivity Disorder (ADHD) affected boys from three to nine times more than girls. Gender-related differences were observed also in some KiTAP subtests: girls had faster reaction times but were less accurate than boys [33]. A study [38] revealed that attention problems in boys were related to less well-developed expressive language skills, while in girls there was a trend for attention problems to be related to lower performance on academic skills. Other studies have found that girls showed a better performance and a higher level of inhibitory control than boys [36]. Gender differences were present also with regard to the speed/arousal dimension, with boys performing faster than girls, both groups aged 6 to 13 [39]. It can be noted that the gender-based differences in attentional performances are still not clear. Austrian male children (6–10 years) were faster in alertness, divided attention, and inhibition, while girls demonstrated better accuracy in flexibility and inhibition. In contrast, in Mexican children (6–12 years) no gender differences in attention and impulse control were identified [33]. This matter is still subject to ongoing debate. The literature on gender difference is scarce and mostly related to clinical populations. In this study, we want to see if the situation in healthy children is similar or dissimilar compared with the clinical setting.

#### 1.2.3. Role of Family Factors

The role of family influences on preschool and school-age cognitive development has received considerable empirical attention from cognitive developmental psychology researchers in the last few decades [40]. The literature shows that a family’s socioeconomic status (SES) could influence the child’s attention performance in early infancy, with low-SES infants showing higher inattention than their high-SES peers already at 6, 9, and 12 months and being less likely to modulate their cognitive skills with respect to stimulus complexity [40]. The link between family socioeconomic status and development trajectories has gained various explanations, with evidence suggesting that higher family incomes are related to more stimulating learning environments. Parents who have financial resources invest in obtaining equipment (toys and books) and in undertaking activities (reading books and teaching abilities) that foster cognitive skills, language development, behavioural functioning, and socioemotional competence skills of their children [41]. It has been speculated that the underlying physiological mechanism leading to reduced attention in low-SES infants could be due to maternal nutrition choices for children [42], which have been found to be associated with long-term alterations in brain development. Moreover, effects of socioeconomic disadvantage on neural structures were observed in 10-year-old children showing widespread modifications in various brain regions and smaller volumes of grey matter. Though the link between parental SES and children’s brain structures could be mediated by other factors (i.e., genetic inheritance), there is considerable evidence for direct environmental effects in other species [43].

Recently, a meta-analysis [44] showed how SES disparities played a relevant role on the executive function performance of children. Families with a low cultural level and income compared to average/high ones showed a considerably higher presence of ADHD. The importance of maternal education for children’s academic outcomes was widely recognized [45]. The number of siblings did not appear to limit children’s cognitive development during early childhood [46]. The literature has emphasized that a good growth environment, with adequate stimulation, facilitates the development not only of attention but of all the most important cognitive abilities [47]. Confirming this, there were data from studies that found a strong correlation between children with ADHD and a low sociocultural and economic family condition [48].

### 1.3. Gap in the Literature: Attention in Healthy Children 

A review of the literature on attention in children revealed few studies that specifically investigated the development and characteristics of the attentional mechanisms of healthy children during the primary school period. Furthermore, developmental studies on children’s attentional skills have been limited to clinical targets, such as children with difficulty or disturbance of attention [49,50,51]. Research on healthy school-age populations which examines the impact of various functions and components of attention on academic achievement is a relatively new research area [16,52]. Within this framework, the present study aimed to provide empirical evidence for the contribution of distinct attention functions in primary school children adopting a friendly computerized system, KiTAP. Research with KiTAP in typically developing school populations is limited to only two developmental cross-cultural studies [36] and one on attentional performance and scholastic achievement [53]. KiTAP has been proved to be an age-sensitive instrument in primary school children (6–11 years). Previous findings suggest that less-complex functions like alertness and sustained and selective attention show early emergence (at kindergarten age) in the course of development and stabilize around the age of 10, while components of executive functions (flexibility, divided attention, inhibitory control) show improvements beyond childhood that continue until early adolescence.

### 1.4. Research Goals


The main objective of this study was to identify children with attentional deficit attending primary school, comparing their scores with the Italian relative norms. The neuropsychological model will guide this research strand, and we expected that time variability in a go ⁄ no-go task, followed by number of errors in a divided attention task and response time variability in an alertness task, could be identified as possible good measures to discriminate between children with and without attention difficulties [10].We expected that attentional functions could improve with growing age, when the child is attending the fourth and fifth grade levels [54,55]. Influence of age on attentional functions could depend on the types of examined functions, with superior executive tasks (i.e., flexibility, divided attention, inhibitory control) improving by increasing age, while other lower tasks (i.e., alertness, sustained and selective attention) remain stable [35].We expected to find gender differences in attentional functions [36,37]. In fact, some studies have revealed that boys, for constitutional reasons, were less likely than girls to stay focused, firm, and alert, while they had faster reaction times [33]. We supposed that boys were faster than girls in the majority of attentive tasks, but that girls were more accurate and precise in their attentional performance.We wanted to verify whether children’s family SES could influence attentional functions, specifically if SES disparities could influence attentive performance among children [44]. We expected that children with lower SES and parental schooling could have difficulties in their attentional functioning.We aimed to understand if the presence of siblings or parents’ level of education could influence the quality of the children’s attentional performance [45,46]. We supposed that children with siblings could have a better attentional performance than children without siblings.


## 2. Materials and Methods

### 2.1. Participants

The participants were 105 children aged 6–10 years old with a mean age of 8.6 (*SD* = 1.04), 57 of whom were female, attending three primary schools in a northeast region of Italy, from the second year of school to the fifth/last year. We received a valid consent form from 115 families among 132 contacted (response rate 87.12%). Ten children were not reached in the assessments because of logistical problems (teachers’ other priorities, ill children in the data collection period, no quiet room available for assessments). Table 1 shows sociodemographic information for the participants, and Table 2 shows family sociodemographic characteristics.

### 2.2. Procedure

The project was successfully proposed to the director of the school, who showed it to the institute councils. A letter explaining the research project was sent to families of students attending the second to the fifth grade, requesting the students’ participation in the study through an attached informed consent form. The inclusion criteria were no history of chronic illness or injury and absence of sensory deficiencies and other pathological aspects. First-class children were not involved as the trial might be too tiring for them, especially for the data collection period that was just at the beginning of the school year (from October to December).

Out of more than 500 letters sent, 132 responses were received with permission to participate, but 17 children were excluded because the informed consent had been signed by just one parent. From these families, 74 filled in the sociodemographic survey and 105 children completed the attention assessment test. 

Students were met individually in a silent and empty room where the laptop with KiTAP for the assessment was located. Each student was assessed in 6 of the full battery of 8 tests. Administering the entire battery would have meant asking the child to be engaged for almost an hour and a half. This was deemed an excessive length as well as having a negative effect on the quality of the child’s classroom performance; additionally, scheduling within the regular school day would have been difficult. Thus, vigilance and visual scanning subtests were removed from the test.

At the end of the test, the psychologist always thanked the participant, stressing the importance of his or her contribution. Overall, the administration lasted 30 min for the oldest and fastest students, and 45 min for the younger ones.

Scores obtained from each subject in each test were stored automatically. They were placed in a table that provided information about the subject, the examiner, and reaction times (RTs) for each trial. In addition, there was a list of results with data about the individual parameters: mean, median, and standard deviation of RT; number of correct and incorrect reactions; and number of omissions. Scores were expressed in percentiles or in T points. The program also offered graphs.

### 2.3. Instruments

#### 2.3.1. KiTAP 

This test has been created to ensure optimal motivation for children during attention testing by providing a design suitable especially for younger children. By increasing motivation and compliance, the validity of the test should be maximised.

Great importance has been attributed to the attentional functions of school-age children. Assessing attention in schoolchildren is crucial for several different diagnostic questions. Yet there is a current lack of test instruments specifically designed to provide a differential measure of young schoolchildren’s attentional abilities.

The battery Test of Attentional Performance for Children KiTAP [32,49,54,56,57] has been constructed with particular attention to the same consideration that was applied in the adult version of the test (TAP). The choice of KiTAP’s tests has been based on the analysis of data from 148 children between the ages of 6 and 10 years tested with TAP. A factor analysis of the data has revealed a factor structure with five independent aspects or factors, which have been represented by a TAP subtest. Table 3 shows KiTAP’s parameters [49].

***Alertness*** (“the witch”) is a central aspect of attentional intensity. Intrinsic alertness is measured with a simple reaction task. In this test, a witch appears at a window and should be driven away as fast as possible by pressing the key. The median provides information on processing speed, while standard deviation indicates the level of stable, maintained alertness. In addition, comparing the performance of tested children with the KiTAP normative performance values, the percentile median and the standard deviation (percentile) were calculated. 

***Distractibility*** (“the sad and the happy ghost”): One of the fundamental aspects of focused attention is the ability to intentionally maintain control over the focus of attention in complex situations and under distracting conditions. Younger children stand out because of their high level of distractibility, through which they frequently lose sight of their goals from one moment to the next when something else captures their attention. A low degree of distractibility is therefore an important prerequisite for concentrated work and is of particular importance for school-aged children. The purpose of this test is to perform a centrally presented decision task, while in half the trials a distracting stimulus appears in the periphery of the visual field. The central stimulus, a cheerful or sad ghost, is designed so that the distinction between cheerful and sad is only possible by focusing visually. The assessed parameters are number of omissions and false alarms: the first indicates the degree of distractibility of the subject, and the second indicates when he or she reacted according to a “suspicion” and not for having really recognized the stimulus. In addition, the two parameters were considered as percentiles, so that we could compare our sample with norms. Scores were considered both in a distractor state and a no-distractor state.

***Divided attention*** (“the owls”): A common experience in daily life is that of paying attention to a number of things or events at once. This requires the ability to divide attention between simultaneously occurring events. In this test, a sequence of acoustic and visual stimuli has to be observed simultaneously and responses to stimuli are made by pressing a key. One sees an owl sitting in a window, which closes its eyes from time to time. This change should be reacted to. Simultaneously, two owls calling to each other can be heard in the background. The number of omissions and median reaction times and false alarms, both for acoustic stimuli and visual stimuli, were measured. The number of omissions is the most important parameter, as it indicates the ability to divert attention from different tasks.

***Flexibility*** (“the dragons’ house”): Selective attention refers not only to the ability to direct attention toward single events and stimuli, but also to redirect attentional focus according to current demands of a situation. The term “flexibility” is used to refer to the ability to intentionally regulate and redirect attention focus. In this test, two dragons of different colour (green and blue) appear to the left and right of the centre of the monitor (a gate) simultaneously. The target stimuli alternate: to begin with, a key has to be pressed on the side at which the green dragon appears. At the next presentation, a key has to be pressed on the side at which the blue dragon appears. The number of false alarms committed and the median of reaction times are the parameters considered, and respective percentiles are calculated for a comparison with the KiTAP normative performance scores.

***Go/no-go*** (“the bat”): Attention comprises not only the control processes through which we take in information from the environment, but equally the control of our reactions and of our behaviour and inhibitory control. This includes the decision as to whether and how we should react as well as the continual, e.g., visuo-motor, control of actions. One of the fundamental processes in this connection is control of impulsive behaviour, that is, ability to suppress an inappropriate reaction. The simplest way to measure impulsive reaction tendencies is by means of the so-called “go/no-go” task. In this test, one sees either a vampire bat or a cat, whereas only the bat should be reacted to. The number of false alarms indicates the ability to inhibit the reaction and the mean of reaction times, which indicates the speed of decision-making ability. In addition, to compare the number of alerts made by our sample with those made by the normative sample, the number of percentile errors was considered.

***Sustained attention*** (“the ghost’s ball”): In this task, the effortful maintenance of selective attention over a longer span of time is tested. In contrast to vigilance, where performance requires the detection of infrequent stimuli that are hard to discriminate and are presented under experimental conditions of extreme monotony, demands with sustained attention are more complex. Conditions of sustained attention or concentration are more characteristic of daily life demands. This task requires comparing a stimulus with a subsequent stimulus to determine whether these two stimuli have a predetermined stimulus feature in common. Stimuli to be compared are ghosts of different colour that appear consecutively at different windows of a castle ruin. This procedure places demands on working memory and flexibility, and in a more complex variant, on the ability to divide attention, since two of the stimulus aspects have to be observed. Parameters are the number of omissions, which indicates the performance stability, and false alarms made, specifically for the first 5 min of the test, for the second 5 min, and for the total test. For the latter condition, the number of omissions and percentile errors have also been considered, so that a comparison with the normative sample of KiTAP could be made. 

#### 2.3.2. Sociodemographic Information 

Parental education and occupational status were measured, by collecting data on education (number of years of school achievement), type and average hours of job, and economic status.

### 2.4. Statistical Analyses Plan

Data were preliminarily checked for normality, adopting the Kolmagorov–Smirnov and Shapiro–Wilks tests. Data distribution was normal, so we decided to use parametric statistics. To answer the research questions, we ran preliminary Pearson’s correlations and ANOVAs to identify the possible significant associations between our variables, adopting post hoc Bonferroni correction if necessary. Perceived economic condition, number of siblings, and parental education level were not inserted in the model because they did not obtain significant associations. Then a series of hierarchical regression analyses were run with school grade (grades 2, 3, 4, and 5), gender (1 = male, 2 = female), and child’s current age as independent variables. The scores obtained at the six individual KiTAP tests (mean reaction times and SD, number of missions, and false alarms) were entered as dependent variables, one by one, choosing the parameters considered as the most significant in the test manual. We will show only the significant obtained results. The interaction factors, if significant, will be indicated.

## 3. Results 

For each KiTAP test, the Italian normative scores for the individual parameters were shown in the manual. These norms were given as percentiles. We assessed the distribution of children along these percentiles, comparing the scores obtained in each subtest with those from Italian standardized norms (Table 4), except for divided attention, where there are no normative values. 

Observing Table 4, we see how a great proportion of children fell in the lower level of percentile categories in their scoring of false alarms and omissions in almost all the attentional tasks. Only medians for distractibility and rapidity assessed by reaction times attested to normal or superior scores, even if exclusively in distractibility and flexibility subtests.

We also ran ANOVAs with perceived economic condition as an independent variable and the several attentive scores as dependent variables inserted one by one. We obtained no significant means differences.

We also ran Pearson’s correlations to identify the significant associations of our independent variables with the several attentional subtests’ parameters. Table 5 shows these correlations.

From this analysis, we identified the variables to insert in the next regression models: gender, school grade, and child’s current age as independent variables. Table 6 shows summary hierarchical regression results.

### 3.1. Alertness

We obtained no significant predictors of alertness median, reaction times, and standard deviation.

### 3.2. Distractibility

For the first condition, with presence of the distractor on the screen, the significant model (R^2^ = 0.13; F_3_ = 5.33; *p* = 0.002) identified female gender (ß = 0.213; *p* = 0.014) as the factor influencing the increase of distractibility RT median. On the other hand, female gender (R^2^ = 0.15; F_3_ = 6.23; *p* = 0.001) impacted as a protective factor in making false alarms (ß = −0.33; *p* = 0.001). 

For the second condition, without distractor, the significant model (R^2^ = 0.19; F_3_ = 7.88; *p* = 0.0001) showed that the distractibility RT median increased by female gender (ß = 0.38; *p* = 0.0001). Another hierarchical model (R^2^ = 0.15; F_3_ = 6.23; *p* = 0.001) identified gender (ß = −0.33; *p* = 0.0001) as the variable influencing the false alarms frequency, more frequently made in boys than girls. 

### 3.3. Divided Attention

Considering the condition of acoustic stimuli, the significant model (R^2^ = 0.07; F_3_ = 2.64; *p* = 0.05) identified the school grade (ß = −0.43; *p* = 0.03) as the factor influencing median RT in the divided attention test. By increasing the child’s school grade, median RT was lower, with better rapidity.

Considering the condition of visual stimuli, the significant model (R^2^ = 0.21; F_3_ = 9.24; *p* = 0.0001) identified female gender (ß = −0.17; *p* = 0.05) and higher child’s school grade (ß = −0.47; *p* = 0.01) as predictors of lower median RT in the divided attention test. A different number of omissions (R^2^ = 0.10; F_3_ = 3.97; *p* = 0.01) resulted from the child’s school grade (ß = −0.55; *p* = 0.008). 

### 3.4. Go/No-Go

Gender (ß = −0.33; *p* = 0.001) influenced significantly the number of false alarms (R^2^ = 0.12; F_3_ = 4.51; *p* = 0.005). The same result was shown for omissions (R^2^ = 0.07; F_3_ = 2.6; *p* = 0.05), with gender impacting significantly (ß = −0.24; *p* = 0.015). Girls had the best performance. 

### 3.5. Flexibility

A series of hierarchical regression analyses was performed, with child’s gender, current age, and school grade as independent variables and RT median, RT median in percentiles, false alarms, and false alarms in percentiles as dependent ones, inserted one by one. Results showed that the number of false alarms and RT median, both in raw score and in percentiles, showed no significant change along these demographic factors.

### 3.6. Sustained Attention

In the first 5 min of testing, the child’s school grade (ß = −0.49; *p* = 0.001) significantly impacted the number of omissions (R^2^ = 0.18; F_3_ = 7.34; *p* = 0.01). Another regression model (R^2^ = 0.26; F_3_ = 11.83; *p* = 0.0001) identified the child’s school grade (ß = −5; *p* = 0.009) as a significant factor influencing the RT median. The children in higher grades showed lower RT medians and made fewer omissions.

For the second part of the test, i.e., the last 5 min, the regression model (R^2^ = 0.07; F_3_ = 2.84; *p* = 0.04) identified female gender (ß = −0.23; *p* = 0.01) as a significant factor impacting a reduced number of false alarms. Omissions were influenced significantly by the child’s school age (ß = −0.59; *p* = 0.003) in another regression model (R^2^ = 0.24; F_3_ = 10.61; *p* = 0.0001), where children with higher school achievement that obtain a better performance.

## 4. Discussion

To answer the first research question, we showed the distribution of children’s performance in standardized tests for the Italian population by percentile categories. A great proportion of children enter the lower level of percentiles of false alarms and omissions in almost all the attentional tasks, compared with Italian norms. On the other hand, reaction time medians that correspond to process rapidity are quite constant, except for alertness and sustained attention tasks, where at least a third or more do not reach the 50th percentile, under the normative cut-off. In the distractibility test, children obtain good scores in rapidity, but at the expense of accuracy, with a higher frequency of omissions of the target stimuli. We know that response inhibition tasks load mainly on central executive measures, predicting reading ability [21], so high frequencies of false alarms and omissions in the go/no-go test could be precursors of reading difficulties in children.

Dealing with the second research question, the results show that for all six KiTAP tests school grade appears to be a key factor, indicating that it significantly influences the performance of students throughout the battery. Their performance along the three school grade groups varies and differs, especially for children in grade 2. Probably from the age of 8 years, there is a transition from an immature phase to a more competent one, as indicated in previous studies [21,35]. An important increase in attentional functions performance is obtained from students in grade 3, and it continues to get better in the last two years of primary school (9–10 years of age). These results identify the school grade as a key factor, also controlling for chronological age, showing how the academic experience and learning through several school cycles are even more important than the chronological age, confirming the important function of schooling achievement in improving superior attention functions [38]. In the alertness test, no significant difference results are found along school grade, chronological age, or gender. The performance is stable throughout the several sociodemographic factors, as found in other studies adopting KiTAP [35,56]. Summarizing, we can state that the division of the sample into the three school grade groups is interesting, because it allows us to observe the worse performances in pupils attending grade 2 compared to the higher classes. Chronological age is not significant in itself, but only associated with academic level. This is an innovative result, obtained with this design, that weights the several possible predictors identifying the best. Probably chronological age is important, but only if accompanied by the schooling experience and activities. Unfortunately, a limit of this study is given by the fact that we did not assess children attending grade 1. Further studies should take into consideration performance among first-graders, enabling researchers to reach a more complete description of the development of attentional functions in childhood.

Possible gender differences [36,37] were also investigated in the third research question. In the distractibility test, rapidity in reaction times is mostly obtained by boys, even if they commit more false alarms. This suggests that boys are faster but less accurate and more focused on their task than are girls, both in the condition with and without the distractor on the screen.

Also, in the divided attention test children attending the last grades show the best performance, while the worst is still for 7-year-old boys attending grade 2: they globally commit more omissions, especially when the target stimulus is visual. Girls in the lower grades show higher reaction times. Perhaps a visual target elicits more attention than an acoustic one on the screen. So, 7-year-old boys show the worst ability to stay concentrated and focused on multiple tasks. When the stimulus is acoustic, the median RT is higher in children belonging to lower grades.

In the go/no-go test, boys show more false alarms and omissions than do girls; probably they press the button less often and then make fewer mistakes.

In the flexibility test, there are no significant risk factors influencing the children’s performance, but in general it is possible to note that children are really fast, but not accurate.

With regard to the last test, sustained attention, in the first 5 min pupils in the higher grades have the best performance, doing the fewest number of omissions and having more rapid reactions (median RT), while the worst performances are by those in the lower grades. Boys committed more false alarms in the last 5 min. This can be explained by the nature of the test: simple and particularly monotonous, the worst performances of pupils can be caused by fatigue, especially in younger children (7 years), and boredom, especially in boys, who have more difficulty in staying focused on the test.

Summarizing, the analyses conducted on the scores obtained from our sample, consisting of 48 boys and 57 girls, show that, overall, the worst performances are obtained from boys for accuracy. Comparing the performance of boys and girls through the three school grade groups, it is to be noted that the number of omissions or false alarms were generally higher for boys, specifically in the go/no-go and distractibility tests, while median reaction times are reduced. In KiTAP trials, therefore, girls generally have better results than boys, showing that their performances are consistently better in accuracy, even if less rapid. Boys are faster but less accurate. Girls in the lower grades have more median RT in the divided attention test with the visual target on the screen.

The third research question aimed to investigate whether the socioeconomic context of the pupil’s family could influence his or her attentional performance [44]. The economic condition is not a factor that appears as a significant variable for attentional performance.

The fourth question involved siblings: Could being a single child or having siblings affect the quality of the attentional performance? The assumption is that the presence of siblings is an important resource of rich social, emotional, and cognitive stimuli [45,46]. Results from this study show that this factor does not affect performance in favour of having siblings, but it is necessary to consider that in our sample of 105 pupils, 78 had siblings and only 27 were single children, so it is difficult to exclude this factor.

### Strengths and Limitations

A strength of this study is the opportunity to investigate attention in an exclusive and authentic way, directly on children in their school setting. The choice of KiTAP as the assessment instrument is valuable from different points of view: it is structured through very simple and immediate tests; it is a computer-based test battery comprising various subtests that cover a broad range of neuropsychological functions of attention; and it allows a good investigation of attention and its mechanisms, even with young or inexperienced children, considering that the different subtests are independent from language. In addition, because it is a computer test presented as a form of play and with fantastic stories and fun, colourful graphics, it is bound to motivate children.

The ample number of participants involved in the project, all from the same geographical area, is a point of strength even if similar residence can also be considered as a limitation; future research should aim to involve other primary schools in other areas to have a sample more representative of the entire country. Another limitation is represented by the inhomogeneity of the sample along age groups, with younger children (6–7 years old) less numerous than older ones (8–10 years old). It will be important to increase the number of participants in the first age group to have a more homogeneous distribution according to age. We also could not have a composite SES to answer our research question because our variable was assessed along different parameters: perceived socioeconomic condition, parental educational level, parental workload in a week. We could not composite these different scales and variables in a unique dimension. Another limitation is that this study could not present the entire KiTAP.

Future research could also focus on better understanding how family socioeconomic condition affects children’s abilities, and even more, to understand whether parents’ minor presence in the lives of children affects the quality of their attentional performance. The presence of siblings does not seem to help the child to reach better attentional functioning. It would be interesting to better understand this phenomenon, also assessing siblings’ attentional functions or observing the sibling relationship during daily family life. Longitudinal studies could be more informative on the development of attentional functioning in children throughout the different school grades, and future studies have to focus on this type of design. Other factors such as the student–teacher relationship and temperament should be taken into consideration in future studies as possible factors that may also play a role in children’s attentional performance.

## 5. Conclusions

Higher school grade matches better performance, especially in advanced attentional tasks, such as divided and sustained attention, with pupils attending the lower grades showing the worst performance independently from their chronological age. Observing these results, we can imagine how school activities and attendance impact upon advanced attentive tasks. Schooling achievement implements and empowers children’s attentive tasks more than the normal advancement of age, providing a sort of training in attentional functions. In this study, we confirm the distinction between basic and nonbasic skills: for the basic tests (i.e., alertness, distractibility, go/no-go), the performance of lower-grade pupils is at least similar to those of their higher-grade companions, while in the nonbasic skills (divided and sustained attention) the child’s scholastic achievement becomes a key factor in ameliorating the child’s performance.

The results show that girls obtained a consistently higher performance throughout the three age groups, especially in the go/no-go and distractibility tests. On the other hand, boys reported best performance with regard to rapidity, even if they are less accurate. Possible educative programmes should focus on amelioration of boys’ inhibitory activity and empowerment of girls’ fast reaction times.

Family factors such as presence of siblings, parental schooling years, and socioeconomic condition do not emerge as possible significant variables on attentional performance. Further studies with a more homogeneous sample of these variables might better investigate these aspects, for example, if the presence of siblings helps in attentive functions.

## Figures and Tables

**Figure 1 behavsci-09-00007-f001:**
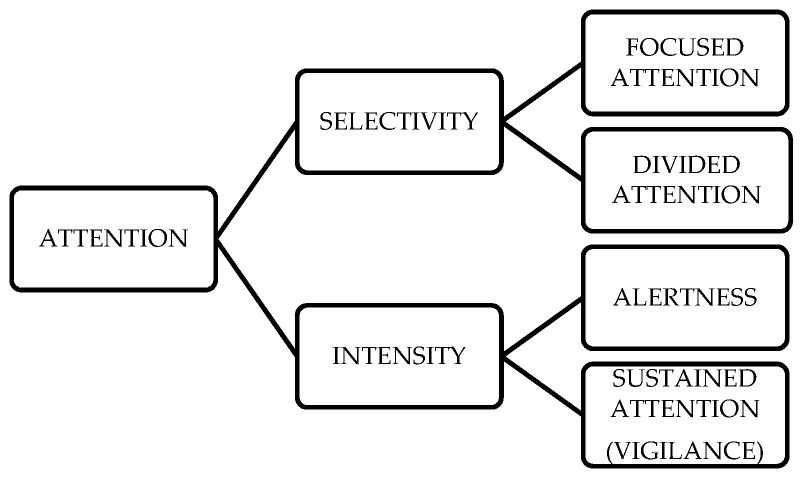
Supervisory attentional control (SAC), Van Zomeren and Brouwer’s Basic Process Sketch, 1994 [6], (in [7], p. 184)

**Table 1 behavsci-09-00007-t001:** Participants’ sociodemographic characteristics.

Class	N	Males	Females
2nd	19 (18.1%)	5	14
3rd	42 (40%)	21	21
4th	27 (25.7%)	13	14
5th	17 (16.2%)	9	8
Total	105 (100%)	48	57

**Table 2 behavsci-09-00007-t002:** Family’s sociodemographic characteristics.

**Parents’ Characteristics**	**Categories**	**Mothers**	**Fathers**
		Frequency	Frequency
Education (years of schooling)	5 years	1%	1.1%
	8 years	26.8%	35.1%
	13 years	60.8%	53.2%
	16 years	5.2%	2.1%
	18 years	5.2%	7.4%
	>18 years	1%	1.1%
Employment	Looking for a job	19.7%	3.1%
	Part-time	50%	3.1%
	Full-time	30.3%	93.8%
Weekly job hours	50 or more	1.6%	17.6%
	40–49	16.1%	61.5%
	30–39	37.1%	18.7%
	20–29	38.7%	2.2%
	10–19	3.2%	0%
	0–9	3.2%	0%
		Mean (SD)	Mean (SD)
Current age		40.11 (4.36)	43.01 (4.56)
**Family’s Characteristics**	**Categories**	**Family**	
		Frequency	
Relationship status	Married	89.1%	
	Divorced/Separated	5.5%	
	Cohabitant	5.4%	
	Single	0%	
Economic situation perceived	Low	21.9%	
	Medium	63.5%	
	High	14.6%	
Home situation	Rent	6.1%	
	In-progress mortgage	45.9%	
	Finished mortgage	37.8%	
	Other	10.2%	
	Range	Mean (SD)	
N familiars	2–6	3.9 (0.75)	
N siblings	1–3	1.2 (0.49)	

**Table 3 behavsci-09-00007-t003:** Parameters in each KiTAP subtest.

Test	Execution Time	Parameters
Alertness	1.5 min	Reaction times (RT): mean, median, standard deviation
Distractibility	3 min	RT median, omissions, false alarms
Divided attention	4.5 min	RT median, omissions, false alarms
Flexibility	1.5–2 min	RT median and RT median in percentiles, false alarms and false alarms in percentiles
Go/no-go	2.5 min	RT median, omissions, false alarms
Sustained attention	10 min	RT median, omissions, false alarms

**Table 4 behavsci-09-00007-t004:** Distribution of children’s performance in standardized tests by percentile categories.

Test		<25°	25°–49°	50°–75°	>75°
Alertness	RT median RT SD	21 32	16 19	31 27	37 27
Distractibility	RT median omissions False alarms	3 66 3	4 23 10	14 13 22	84 3 70
Flexibility	RT median False alarms	6 28	14 41	27 14	59 34
Go/no-Go	RT median Omissions False alarms	9 13 33	20 77 38	31 5 18	45 10 15
Sustained attention	RT median omissions False alarms	14 31 46	21 26 30	26 28 19	44 20 10

**Table 5 behavsci-09-00007-t005:** Pearson’s correlations between the examined variables and each attentional functioning scale.

	No/Yes Presence of Sibling	Child’s Current Age	Mother’s Schooling Years	Father’s Schooling Years	Child’s Gender (1 = Male, 2 = Female)
RT Median Alertness	r = −0.12; *p* = 0.28	r = −0.41 **; *p* = 0.001	r = 0.05; *p* = 0.63	r = −0.06; *p* = 0.52	r = 0.21 * *p* = 0.02
SD Alertness	r = −0.13; *p* = 0.25	r = −0.29 **; *p* = 0.003	r = 0.02; *p* = 0.82	r = −0.11; *p* = 0.27	r = −0.03; *p* = 0.76
Omissions Distractibility_Total	r = 0.01; *p* = 0.89	r = −0.17; *p* = 0.08	r = 0.14; *p* = 0.16	r = 0.20; *p* = 0.05	r = −0.11; *p* = 0.27
False Alarms Distractibility_Total	r = −0.05; *p* = 0.65	r = −0.24 *; *p* = 0.01	r = −0.14; *p* = 0.16	r = −0.09; *p* = 0.93	r = −0.23 *; *p* = 0.01
RT Median Distractibility_Total	r = 0.12; *p* = 0.30	r = −0.18; *p* = 0.05	r = 0.05; *p* = 0.58	r = −0.08 *p* = 0.43	r = 0.35 **; *p* = 0.0001
False Alarms Flexibility	r = −0.09; *p* = 0.42	r = −0.15; *p* = 0.11	r = −0.19; *p* = 0.05	r = 0.07; *p* = 0.49	r = −0.13; *p* = 0.16
RT Median Flexibility	r = −0.15; *p* = 0.19	r = −0.37 **; *p* = 0.0001	r = −0.07; *p* = 0.49	r = −0.08; *p* = 0.40	r = 0.16; *p* = 0.08
Omissions Go/No-Go	r = 0.05; *p* = 0.60	r = −0.12; *p* = 0.20	r = −0.06; *p* = 0.57	r = 0.03; *p* = 0.77	r = −0.22 *; *p* = 0.02
False alarms Go/No-Go	r = −0.13; *p* = 0.24	r = −0.1; *p* = 0.32	r = −0.09; *p* = 0.38	r = −0.09; *p* = 0.36	r = −0.31 **; *p* = 0.001
RT Median Go/ No-Go	r = 0.20; *p* = 0.07	r = −0.42 **; *p* = 0.0001	r = 0.19; *p* = 0.07	r = −0.12; *p* = 0.25	r = 0.18; *p* = 0.06
Omissions Sustained_attention_Total	r = −0.08; *p* = 0.44	r = −0.41 **; *p* = 0.0001	r = −0.18; *p* = 0.06	r = 0.05; *p* = 0.65	r = −0.1; *p* = 0.31
False Alarms Sustained attention_Total	r = −0.04; *p* = 0.68	r = −0.09; *p* = 0.34	r = −0.03; *p* = 0.79	r = −0.01; *p* = 0.91	r = −0.20 *; *p* = 0.04
RT Median Sustained attention_Total	r = −0.13; *p* = 0.25	r = −0.47 **; *p* = 0.0001	r = −0.03; *p* = 0.79	r = 0.01; *p* = 0.87	r = 0.12; *p* = 0.20
Omissions Divided_attention_acoustic	r = −0.08; *p* = 0.48	r = −0.22 *; *p* = 0.02	r = −0.19; *p* = 0.05	r = −0.12; *p* = 0.25	r = −0.03; *p* = 0.73
False Alarms Divided attention_acoustic	r = −0.10; *p* = 0.36	r = 0.04; *p* = 0.67	r = −0.23 *; *p* = 0.02	r = −0.16; *p* = 0.10	r = −0.11; *p* = 0.25
RT Median Divided attention_acoustic	r = −0.19; *p* = 0.08	r = −0.18; *p* = 0.07	r = −0.04; *p* = 0.65	r = −0.05; *p* = 0.59	r = −0.006; *p* = 0.95
Omissions Divided_attention_visual	r = 0.08; *p* = 0.43	r = 0.02; *p* = 0.80	r = 0.09; *p* = 0.37	r = 0.09; *p* = 0.37	r = −0.55; *p* = 0.57
False Alarms Divided attention_visual	r = 0.02; *p* = 0.80	r = −0.12; *p* = 0.23	r = −0.15; *p* = 0.14	r = −0.15; *p* = 0.13	r = −0.79; *p* = 0.42
RT Median Divided attention_visual	r = −0.01; *p* = 0.99	r = −0.38 **; *p* = 0.0001	r = 0.07; *p* = 0.49	r = 0.18; *p* = 0.08	r = −0.10; *p* = 0.28

* *p* < 0.05; ** *p* < 0.01.

**Table 6 behavsci-09-00007-t006:** Summary of hierarchical regression results.

RESULTS	Alertness	Distractibility	Divided Attention	Go/no-go	Flexibility	Sustained Attention
School grade (3 levels: 2, 3, 4–5)	NS	NS	*p* < 0.05 omissions visual stimuli conditions *p* < 0.05 RT median visual stimuli conditions *p* < 0.05 RT median acoustic stimuli conditions	NS	NS	*p* < 0.05 for number of omissions and RT median (first 5 min) omissions (second 5 min)
Gender (male/female)	NS	*p* < 0.05 RT median and false alarms (with and without distractor conditions)	*p* < 0.05 RT median visual stimuli condition	*p* < 0.05 false alarms and omissions	NS	*p* < 0.05 for number of false alarms (second 5 min) (second 5 min and total time)

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
