# Peer review of "Which Factors Influence Attentional Functions? Attention Assessed by KiTAP in 105 6-to-10-Year-Old Children"

_behavsci, 2019, doi:10.3390/bs9010007_

Round 1

Reviewer 1 Report

Authors presents a valuable examination of factors associated with children’s attention. This is a topic worthy of much study as attention is related to higher order cognitive functions necessary for academic achievement. Although the manuscript has potential, I offer the following suggestions to strengthen the paper for publication and suggest a thorough check for grammatical and sentence structure errors.

Abstract: Given the feedback/comments suggested below, abstract should be updated accordingly.

Introduction Questions/Comments: The justification for Authors’ research questions were unclear. This is mainly due to insufficient background information. The development of attention and its operationalization in this study could be achieved by further elaborating on Van Zomeren & Brouwer’s model. Overall, Authors are urged to elaborate more on the studies incorporated in their introduction. For instance, Authors state that “Attention has been significantly associated with fine motor control from 5 to 11 years old, so a possible delay in attentional performance could influence other important children’s development abilities” and then stop. No development or support other than a citation is provided. This is repeatedly done throughout the manuscript. One last example is when Authors state “consideration should be given to the child’s characteristics, taking into account the influence of many factors [5]: biological characteristics of the child, maturation levels of the central nervous system (CNS), general cognitive and emotional capacity of the child, and, finally, environmental variables, namely his/her personal experiences, context in which he/she lives. School is one of the most significant and privileged developmental context for the child.” What evidence is there to support these statements? Because I am familiar with the literature, I agree with some of these assumptions but Authors need to better argue their point by explaining the empirical evidence supporting their claim.

Line 59 – needs citation

Section 1.2.2 – There are other studies on normal child populations  discussing potential gender differences in attention. For example:

Else-Quest, N. M., Hyde, J. S., Goldsmith, H. H., & Van Hulle, C. A. (2006). Gender differences in temperament: A meta-analysis. Psychological Bulletin, 132(1), 33–72. https://doi.org/10.1037/0033-2909.132.1.33

Zevenbergen, A. A., & Ryan, M. M. (2010). Gender differences in the relationship between attention problems and expressive language and emerging academic skills in preschool-aged children. Early Child Development and Care, 180(10), 1337-1348. doi:10.1080/03004430903059292

KiTAP is also introduced in this section but not explained.

Section 1.2.3 – The same lack of elaboration and support is evident in this section as well. I found it interesting that Authors stated that “attention has been mainly linked to continuous and bidirectional interaction between genes, biological structures and functions on the one hand, and environmental factors on the other” but neglect to ever discuss any biological functions or structure associated with attention.

Section 1.3 – You can’t say “a large set of studies” and then not mention or cite any that corroborate your claim. Moreover, making statements such as “identified as the best measures to discriminate between children with and without attention difficulties” without actually discussing the evidence for such a strong claim should always be avoided. There are many different assessments now available so such a statement needs to be thoroughly argued.

Section 1.4 – In line 132, what do Authors mean by 2 classes? Author’s state that “The literature was divided on this topic: according to some authors, attentional functions improved by increasing age, while, according to others, improvements with age were not significant [22-23]’ but never delve into them to suggest possibilities for their discrepancy an dhow their study different or will help fill a gap.

Eight-year-olds “marked improvement” is never discusses before.

In research question #4, it is important to remember that executive function is not the same as attention. Attention is an important and vital aspect of executive function.

Furthermore, it wasn’t until I read #4 that I started to realize that SES was the focus of the study which means that SES  is underdeveloped in the introduction. Authors need to explain why SES may influence attention. The "WHY's" are consistently omitted in the introduction. It is also important for Authors to really address the significance of their research questions. Why should we care that there may be gender or SES differences?

Method & Results Questions/Comments: Authors discuss shortening their task due potential disengagement, but isn’t that what Authors were interested in; attention? Furthermore, Authors state that “This test has been created to ensure optimal motivation for children during attention testing by providing a design suitable especially for younger children.”lastly, given the model discussed in the introduction, why was vigilance on of the ones removed?

Section 2.3.1 Some of this explanation belongs in earlier sections. Go/no go is not solely an attention task but an inhibitory control task.

How was SES calculated? It seems like only perceived SES was used. Provide reliability and validly SES measure as it is operationalized differently across studies. Was a composite measure of SES tried? 

Due to the numerous analyses was a bonferroni correction applied? If not, many significant results, may not actually be significant.

Provide descriptive statistics and correlations between study key variables. Were t-test performed to even preliminary suggest gender differences?

Were interactions with SES or gender examined? The examination of interactions may help make this this manuscript more nuanced.

Discussion Questions/Comments: Without knowing the answers to some of the questions above, it is hard to appropriate evaluate this section.

Author Response

Authors presents a valuable examination of factors associated with children’s attention. This is a topic worthy of much study as attention is related to higher order cognitive functions necessary for academic achievement. Although the manuscript has potential, I offer the following suggestions to strengthen the paper for publication and suggest a thorough check for grammatical and sentence structure errors.

Abstract: Given the feedback/comments suggested below, abstract should be updated accordingly.

Response: We changed the introduction along your precious suggestions so that also the abstract now is coherent

Introduction Questions/Comments: The justification for Authors’ research questions were unclear. This is mainly due to insufficient background information. The development of attention and its operationalization in this study could be achieved by further elaborating on Van Zomeren & Brouwer’s model. Overall, Authors are urged to elaborate more on the studies incorporated in their introduction. For instance, Authors state that “Attention has been significantly associated with fine motor control from 5 to 11 years old, so a possible delay in attentional performance could influence other important children’s development abilities” and then stop. No development or support other than a citation is provided. This is repeatedly done throughout the manuscript.

Response: We further elaborated the model, giving more background information. See lines 41-49 and lines 51-54, 62-70.

Response: We decided to delete the study on visual-spaced tasks because we didn’t assess this domain in our KITAP battery. We supported more the provided studies/citations. 

One last example is when Authors state “consideration should be given to the child’s characteristics, taking into account the influence of many factors [5]: biological characteristics of the child, maturation levels of the central nervous system (CNS), general cognitive and emotional capacity of the child, and, finally, environmental variables, namely his/her personal experiences, context in which he/she lives. School is one of the most significant and privileged developmental context for the child.” What evidence is there to support these statements? Because I am familiar with the literature, I agree with some of these assumptions but Authors need to better argue their point by explaining the empirical evidence supporting their claim.

Response: We added and discussed more studies in our introduction to sustain these assumptions. See lines 143-154.

Line 59 – needs citation

Response: We added the citation. See line 41

Section 1.2.2 – There are other studies on normal child populations discussing potential gender differences in attention. For example:

Else-Quest, N. M., Hyde, J. S., Goldsmith, H. H., & Van Hulle, C. A. (2006). Gender differences in temperament: A meta-analysis. Psychological Bulletin, 132(1), 33–72. https://doi.org/10.1037/0033-2909.132.1.33

Zevenbergen, A. A., & Ryan, M. M. (2010). Gender differences in the relationship between attention problems and expressive language and emerging academic skills in preschool-aged children. Early Child Development and Care, 180(10), 1337-1348. doi:10.1080/03004430903059292

Response: We inserted the first study at lines 143-144 and the second one at lines 215-218, thank you for the suggestions

KiTAP is also introduced in this section but not explained.

Response: We explained it in depth in the method section. We think that it could be redundant so we only added a briefly explanation in the Introduction section giving some studies adopting this instrument. See lines 62-67.

Section 1.2.3 – The same lack of elaboration and support is evident in this section as well. I found it interesting that Authors stated that “attention has been mainly linked to continuous and bidirectional interaction between genes, biological structures and functions on the one hand, and environmental factors on the other” but neglect to ever discuss any biological functions or structure associated with attention.

Response: We decided to delete this information because we didn’t study these biological functions in our study.

Section 1.3 – You can’t say “a large set of studies” and then not mention or cite any that corroborate your claim.

Response: We deleted this sentence, while we added KITAP studies to sustain our aims. See lines 238-245.

Moreover, making statements such as “identified as the best measures to discriminate between children with and without attention difficulties” without actually discussing the evidence for such a strong claim should always be avoided. There are many different assessments now available so such a statement needs to be thoroughly argued.

Response: Thank you for your suggestions. We decided to formulate this sentence in another way. See lines 248-250.

Section 1.4 – In line 132, what do Authors mean by 2 classes?

Response: We clarified better. See line 256

Author’s state that “The literature was divided on this topic: according to some authors, attentional functions improved by increasing age, while, according to others, improvements with age were not significant [22-23]’ but never delve into them to suggest possibilities for their discrepancy and how their study different or will help fill a gap.

Response: We added more studies about age differences in the section 1.2.1 and reformulated this statement. See lines 159-164 and 184-190.

Eight-year-olds “marked improvement” is never discusses before.

Response: We delated this part, to be more coherent with our analysis

I

n research question #4, it is important to remember that executive function is not the same as attention. Attention is an important and vital aspect of executive function.

Response: We did this correction. See lines 341-343.

Furthermore, it wasn’t until I read #4 that I started to realize that SES was the focus of the study which means that SES is underdeveloped in the introduction. Authors need to explain why SES may influence attention. The "WHY's" are consistently omitted in the introduction. It is also important for Authors to really address the significance of their research questions. Why should we care that there may be gender or SES differences?

Response: studied SES is related to children socio-demographic information (child’s age, gender, scholastic achievement) and these factors were more stressed in the introduction. The perceived economic status seemed to be a key element in developing ADHD, but in our study we didn’t find this significance.

Method & Results Questions/Comments: Authors discuss shortening their task due potential disengagement, but isn’t that what Authors were interested in; attention? Furthermore, Authors state that “This test has been created to ensure optimal motivation for children during attention testing by providing a design suitable especially for younger children.” lastly, given the model discussed in the introduction, why was vigilance on of the ones removed?

Response: We had a limited time in the schools and we were not allowed to stay more than 45 minutes per child, so we had to choose the trials to stay within the established test session time namely 45 minutes. we had to exclude some tasks that would have requested longer evaluations for this reason. We put this explanation also in the main text. See lines 384-385.

Section 2.3.1 Some of this explanation belongs in earlier sections. Go/no go is not solely an attention task but an inhibitory control task.

Response: We corrected it, thank you. See line 184-185 and 411

How was SES calculated? It seems like only perceived SES was used. Provide reliability and validly SES measure as it is operationalized differently across studies. Was a composite measure of SES tried? 

Response: We have only some SES info: parents’ schooling, perceived economic condition and home situation, but we haven’t a composite measure of SES.

Due to the numerous analyses was a bonferroni correction applied? If not, many significant results, may not actually be significant.

Response: The ANOVA analyses didn’t obtain significant results also adopting Bonferroni correction. See lines 493-494 We used a series of regression analysis and in this case the Bonferroni correction isn’t necessary

Provide descriptive statistics and correlations between study key variables. Were t-test performed to even preliminary suggest gender differences?

Response: We decided to run regression analyses to have a model to assess the best independent variables impacting on children’s attentional functions. According to our statisticians’ opinion, it could be confounding to run also t-tests to identify possible gender differences or correlations to our data, because the regression model just gives us these indications, weighing at the same time all the possible predictive factors.

Were interactions with SES or gender examined? The examination of interactions may help make this this manuscript more nuanced.

Response: We examined interactions but they were not significant. We added this information in the analyses plan. See line 502

Reviewer 2 Report

The study is interesting and could have some resonance in the educational field.

Suggestions.

Lines 36-41: Need citations.

Line 52-54 Need more explanations to sustain your idea.

Lines 87-90: Need citations.

Please highlight the novelty of your study according with previous studies.

Lines 215-270 are copy-past from https://www.psytest.net/index.php?page=Ablenkbarkeit_kitap&hl=en_US. I recommend adding the link or to rewrite the content.

I recommend moving the Strengths and limitation in the end part of the Discussion section.

Please add more conclusions according with your specific aims.

Author Response

The study is interesting and could have some resonance in the educational field.

Response: Thank you

Suggestions.

Lines 36-41: Need citations.

Response: We inserted a citation and we added other significative studies. Lines 41-47

Line 52-54 Need more explanations to sustain your idea.

Response: We deleted this part, in accordance with the other reviewer’s opinion. We decided to delete the study on visual-spaced tasks because we didn’t assess this domain with the KITAP battery. We supported more the provided studies/citations. 

Lines 87-90: Need citations.

Response: We added new citations and discussed it more deeply. See lines 143-154

Please highlight the novelty of your study according with previous studies.

Response: See lines 78-81 in the Introduction and lines 592-595 in the Discussion

Lines 215-270 are copy-past from https://www.psytest.net/index.php?page=Ablenkbarkeit_kitap&hl=en_US. I recommend adding the link or to rewrite the content.

Response: We rewrote the sentences and inserted this citation as you suggested us (line 413). Thank you

I recommend moving the Strengths and limitation in the end part of the Discussion section.

Response: We did it.

Please add more conclusions according with your specific aims.

Response: We add more conclusions. Lines 677-679 and lines 686-688.

Round 2

Reviewer 1 Report

Authors revised manuscript addresses most comments and concerns outlined in the first revision. A few points, however, still need to be addressed before publication.

Introduction Questions/Comments:

Intro needs to be better organized. For instance, Section 1 should introduce attention, then its development, the model being followed and its rational or improvement from other theories, KITAP as a measure, and lastly transition into the significance of attention in educational contexts that leads into Section 1.2.

Ln 44: revise the use of “inhabitation”

Ln 64: this seems out of place. The definition should go right after its first mentioned.

Section 1.2 – Ln. 91-92– elaborate on this literature to effectively argue your point.

Section 1.2.1 & .2 – Ln 24-30 has spacing problems. Spacing issues throughout.

These sections need to argue what additional information this study plans to provide and add to the literature.

Section 1.2.3 –Explanation for the mechanism in which SES, parental age, and number siblings may influence child outcomes is needed.

Section 1.3 – Ln. 69-73 - Needs further development. Why? What’s the evidence? Additionally, this would be a great spot to emphasize that KITAP uses these among other tasks to assess attention. As it stands now, it’s seems like go/no go is separate from KITAP.

Section 1.4 – Goal #1 – Should include how the prescribed model will guide this goal.

Goal #2 - inhibitory control is repeated

Goal #3 – Was the hypothesis then that females would outperform males on all areas of attention?   

Goal # 4 & 5 – What were your specific hypotheses?

Information on Ln. 91-95 should be elaborate on and moved to Section 1.2.3

Method & Results Questions/Comments:

Authors did not provide a reason for not attempting an SES composite/latent construct.

Ln 28-29 – It is customary that correlation tables are included to corroborate statements. Provide descriptive statistics and correlations between study key variables.

You are using the same data set to test several hypotheses. Regardless of chronology, each hypothesis test that makes use of a given data set should have some sort of adjustment made to the p-value. You are testing the same data multiple times.

Discussion Questions/Comments:

This was not a longitudinal study so results should be interpreted with this in mind.

Ln 55-57 - Please provide some possible reasoning for this study’s contradictory result compared to other literature.

Ln 66-67 is misleading and should be remove. The fact that authors discuss inhibitory control and executive function contradicts this statement.

Ln 67-72 – Authors support for the use of KiTAP is weak. Supplement these statements (some come across as opinions) with empirical evidence.

Few limitations noted. For example, this study could not provide the entire KiTAP. Other factors such as student-teacher relationship and temperament that may also play a role in children attentional performance were not studied.

A conclusion/summary paragraph should be included.

Author Response

Authors revised manuscript addresses most comments and concerns outlined in the first revision. A few points, however, still need to be addressed before publication.

Introduction Questions/Comments

Intro needs to be better organized. For instance, Section 1 should introduce attention, then its development, the model being followed and its rational or improvement from other theories, KITAP as a measure, and lastly transition into the significance of attention in educational contexts that leads into Section 1.2.

We reorganized the introduction following your suggestions. See lines 32-78.

Ln 44: revise the use of “inhabitation”

Thanks. We have changed it with "response inhibition".

Ln 64: this seems out of place. The definition should go right after its first mentioned.

The definition of Kitap has been replaced. See lines 72-77.

Section 1.2 – Ln. 91-92– elaborate on this literature to effectively argue your point.

We deeply elaborated this literature on lines 94-99 and 105-115.

Section 1.2.1 & .2 – Ln 24-30 has spacing problems. Spacing issues throughout.

Spacing has been controlled throughout the paper (from the first section to the end).

These sections need to argue what additional information this study plans to provide and add to the literature.

We clarified the value added of this study. See lines 150-152 and 165-172

Section 1.2.3 –Explanation for the mechanism in which SES, parental age, and number siblings may influence child outcomes is needed.

Explanation for SES are given on lines 177-192

Section 1.3 – Ln. 69-73 - Needs further development. Why? What’s the evidence? Additionally, this would be a great spot to emphasize that KITAP uses these among other tasks to assess attention. As it stands now, it’s seems like go/no go is separate from KITAP.

KITAP assesses also go/no go task that could develop/increase also during adolescence but it starts in the childhood developmental stage.  We gave more evidence on KITAP literature and attention functions in the above sections, as you suggested to us. See lines 230-232.

Section 1.4 – Goal #1 – Should include how the prescribed model will guide this goal. 

Indications for goal 1 are given on lines 241-245.

Goal #2 - inhibitory control is repeated

We carefully checked and we didn’t find a repetition of inhibitory control in this paragraph

Goal #3 – Was the hypothesis then that females would outperform males on all areas of attention?   

We explained better our hypothesis. See lines 253-245

Goal # 4 & 5 – What were your specific hypotheses?

We specified our hypotheses. See lines 248-249 and lines 251-252

Information on Ln. 91-95 should be elaborate on and moved to Section 1.2.3

We moved this part and we elaborated better this section as you requested us above. See lines 198-202

Method & Results Questions/Comments:

Authors did not provide a reason for not attempting an SES composite/latent construct.

We couldn’t have a composite SES to answer our research question because our variable was assessed along different parameters: perceived socio-economic condition, parental educational level, parental workload in a week. We couldn’t composite these different scales and variables in a unique dimension. We inserted this as a limit. See lines 557-561.

Ln 28-29 – It is customary that correlation tables are included to corroborate statements. Provide descriptive statistics and correlations between study key variables.

We included the correlation tables summed in a global one as you suggested to us. See Table 5 and lines 407-411. We didn’t include descriptive statistics because, in our opinion, it would be too extended and less informative. We added also a part on ANOVAs at lines 404-406.

You are using the same data set to test several hypotheses. Regardless of chronology, each hypothesis test that makes use of a given data set should have some sort of adjustment made to the p-value. You are testing the same data multiple times.

The dependent variables are different in each test and the independent variables taken into consideration are stable factors (gender, child’s current age, child’s school grade) and in the regression models we didn’t test the same data multiple times.

Discussion Questions/Comments:

This was not a longitudinal study so results should be interpreted with this in mind.

We put it in the recommendations for future research. See lines 567-569

Ln 55-57 - Please provide some possible reasoning for this study’s contradictory result compared to other literature.

We explained better this result. See lines 493-495

Ln 66-67 is misleading and should be remove. The fact that authors discuss inhibitory control and executive function contradicts this statement.

We removed this statement. See lines 576-578.

Ln 67-72 – Authors support for the use of KiTAP is weak. Supplement these statements (some come across as opinions) with empirical evidence.

We supported more the KITAP use. See lines 541-548

Few limitations noted. For example, this study could not provide the entire KiTAP. Other factors such as student-teacher relationship and temperament that may also play a role in children attentional performance were not studied.

We added these limitations in the strengths/limits section. Thank you. See lines 625-629 and lines 635-639.

A conclusion/summary paragraph should be included.

We just included it and expanded it a little. See lines 560-561 and 569-571

Reviewer 2 Report

The authors  improved the manuscript according with the suggestions. 

Author Response

Thank you for your suggestions and your work.